

# Demographic and genetic characterization of harvested *Corbicula fluminea* populations

Weikai Wang[1,*], Xiongjun Liu[2,*], Noé Ferreira-Rodríguez[3], Weiwei Sun[1], Yanli Wu[1], Shan Ouyang[1], Chunhua Zhou[1] and Xiaoping Wu[1,2]

[1] School of Life Sciences, Nanchang University, Nanchang, China
[2] Key Laboratory of Poyang Lake Environment and Resource Utilization, Ministry of Education, School of Resources Environmental and Chemical Engineering, Nanchang University, Nanchang, China
[3] Departamento de Ecoloxía e Bioloxía Animal, Universidade de Vigo, Vigo, Spain
[*] These authors contributed equally to this work.

## ABSTRACT

The freshwater clam *Corbicula fluminea s.l.* is an edible freshwater bivalve of economic value in Asia. The species has been particularly well studied in the invaded range. However, there is a lack of knowledge in its native range where it supports an increasing commercial harvest pressure. Among Asiatic countries, China accounts for 70% of known commercial harvest and aquaculture production. We aim to characterize here wild *C. fluminea s.l* populations exposed to commercial harvest pressure in Poyang Lake Basin. We found higher biomass, density and genetic diversity in lake populations compared to peripheral populations (i.e., lake tributaries). Given that lake habitats support more intense harvest pressure than peripheral habitats, we suggest that demographic and genetic differentiation among subpopulations may be influenced in some degree by different harvest pressure. In this regard, additional demographic and/or genetic changes related to increasing harvest pressure may place population at a higher risk of extirpation. Altogether, these results are especially relevant for maintaining populations at or above viable levels and must be considered in order to ensure the sustainability of the resource.

## INTRODUCTION

Sustainable commercial harvest has been prescribed as an effective strategy for the conservation of natural resources (*Struhsaker, 1998*). The common flaw in this conservation approach, is that commercial harvest rarely consider anything but the species being harvested and ignores the complexity of interactions in natural systems (but see *Díaz-Balteiro & Romero, 2004*). In this sense, harvest is usually non-random (size-, sex-, or stage-selective) which has the potential to drive artificial selection on wild populations (*Calvete, Angulo & Estrada, 2005*; *Pelletier, Festa-Bianchet & Jorgenson, 2012*). That is why it is commonly referred to as "unnatural selection" (*sensu Allendorf & Hard, 2009*).

Corresponding authors
Chunhua Zhou,
zhouchunhuajx@hotmail.com
Xiaoping Wu, xpwu@ncu.edu.cn

The main changes associated to unnatural selection are related to demographic and genetic characteristics on wild populations. For example, harvesting large individuals may cause a reduction of biomass, and biases populations towards younger ages and smaller sizes at maturation (*Allendorf et al., 2008*; *Kendall et al., 2014*). In the other side, harvesting smaller individuals may cause reduced population abundance and biomass, and biases populations towards larger sizes at maturation (*Dunlop, Shuter & Dieckmann, 2007*). From the genetic perspective, selective harvest on wild populations has been coupled to loss of genetic diversity (*Harris, Wall & Allendorf, 2002*). Nevertheless, it has also the potential to alter the gene flow among populations, increasing genetic diversity if recruitment is dominated by migrant individuals instead self-recruiting ones (*Miller, Maynard & Mundy, 2009*). In spite of this apparently positive effect, increasing genetic diversity may result in the loss of genes and genotypes associated with the individual performance at the local environment. Hence, to ensure the persistence over time of harvested populations, it is crucial to incorporate demographic and genetic considerations into management.

The freshwater clam *Corbicula fluminea s.l.*, is an edible freshwater bivalve of economic value in Asia (*Liu, Zhang & Wang, 1979*). Part of the native range of *C. fluminea s.l.*, particularly in Eastern China, is characterized by unique shallow lakes with high habitat heterogeneity and surrounded by modified inland artisanal aquaculture systems (*Fu et al., 2003*; *Liu et al., 2019*). Among them, Poyang Lake is a biodiversity hotspot (*Jin et al., 2012*; *Li et al., 2019*). It has 155 known mollusks species, of which more than 50% of the bivalves and gastropods are endemics (*Xiong, Ouyang & Wu, 2012*; *Zhang et al., 2013*). However, due to human activities (e.g., habitat loss and fragmentation, sand dredging, water pollution, overfishing), freshwater biodiversity is in decline on an unprecedented scale (*Jin et al., 2012*; *Huang, Wu & Li, 2013*; *Li et al., 2019*). For example, many unionoid mussels in the Poyang Lake are considered to be threatened (*Shu et al., 2009*; *Xiong, Ouyang & Wu, 2012*; *Zhang et al., 2013*). As a consequence, declining populations of unionoid mussels have provided an empty niche colonized by corbiculid clams (*Xiong, Ouyang & Wu, 2012*; *Zhang et al., 2013*; *Liu et al., 2017*). In addition, *C. fluminea s.l.* has greatly expanded its distribution over the world associated to human activities. Actually, *C. fluminea s.l.* appear to succeed wherever the temperature is not excessively low (i.e., absent from Polar Regions; *Crespo et al., 2015*). Because of the ecological impact, *C. fluminea s.l.* has been long studied in the invaded range (*Cohen et al., 1984*; *Araujo, Moreno & Ramos, 1993*; *Sousa, Antunes & Guilhermino, 2008*); it is in the native range where information about the species is lacking.

The goal of this study was to provide one first demographic and genetic characterization of *C. fluminea* (Müller, 1774) populations in the native distribution range. First, we aimed to test whether lake and peripheral (i.e., lake tributaries) populations show different biomass density parameters. Second, we aimed to test whether (1) genetic diversity will be related to drifting organisms from lake to peripheral populations or (2) genetic diversity will be related to migration from peripheral populations to the main lake. Finally, we discuss wether observed demographic and genetic parameters could be related

with different harvest pressure between lake and peripheral habitats, and how these results could be of utility in the invaded range.

## METHODS

### Commercial harvest and biomass–density relationship

First, commercial harvest data (in tons per year) for the period 1950 to 2017 were obtained from the Food and Agriculture Organization (FAO; http://www.fao.org) of the United Nations food balance sheets (2016). To the best of our knowledge this service constitutes the best available source of information for landings' estimation. Data were aggregated by region (i.e., Asia) and country (i.e., China), fishing area (i.e., inland waters), animal group (i.e., freshwater mollusks) and species (i.e., *Corbicula fluminea*).

Second, we aimed to characterize *C. fluminea* fishery in two zones (lake and peripheral habitats) across six sites (two in lake habitats and four in peripheral habitats). Observations from multiple visits were used to estimate the number of harvest boats in each zone. *Corbicula fluminea* individuals were directly purchased from local fishers between 2018 and 2019. Fishers were identified by their on-site location and obtained clams were bagged, labelled, returned to the laboratory and preserved in 95% ethanol. Following, samples were counted and measured to the nearest 0.1 mm along the longest posterior-anterior axis using Vernier calipers.

In addition, we established a total 84 sampling points covering 28 sampling sections and eight sampling areas, which included (1) lake habitats representative of high harvest pressure, and (2) peripheral habitats representative of low harvest pressure (Fig. 1). Samples were collected at four sampling time points: December 2016 and April, July and October 2017. Two collection methods were used in this study according to habitat features. First, a modified Petersen grab (area of 1/16 m$^2$) was used for sampling in Poyang Lake and the Yangtze River. Second, a quantitative Surber sampler for benthic macroinvertebrates (30 × 30 cm, 500 μm mesh, SN-312, China) was used in mid-size rivers. All samples were collected in triplicate at each sampling site. Retrieved *C. fluminea* samples were transported to the laboratory at the University of Nanchang in a low temperature incubator. Specimens were counted and weighed with an electronic balance (HANGPING FA1204B) to the nearest 0.1 g.

Environmental data for the 28 sampling sections were referenced from *Li et al. (2019)*. Briefly, dissolved oxygen (DO), hydrogen ions (pH), turbidity (TURB), salinity (Sal) and water temperature (T) were measured with a multiparameter probe (YSI 650MDS, YSI Inc. Yellow Springs, OH, USA). Chlorophyll-a (Chl-a) was measured with a chlorophyll meter (HL - 168C06, China), and the water velocity (V) was measured with a velocity meter (FP111, Global Water Instrumentation, TX, USA).

Data were tested for homogeneity of variance using Levene's Test and Kolmogorov–Smirnov's Test for normality. Differences on the size of harvested *C. fluminea* among habitats were tested using Analysis of Variance (ANOVA) on ln (x) transformed data. No significant differences were found in density or biomass between sampling time points (ANOVA, $P > 0.05$; results not shown). Hence, the four sampling time points were
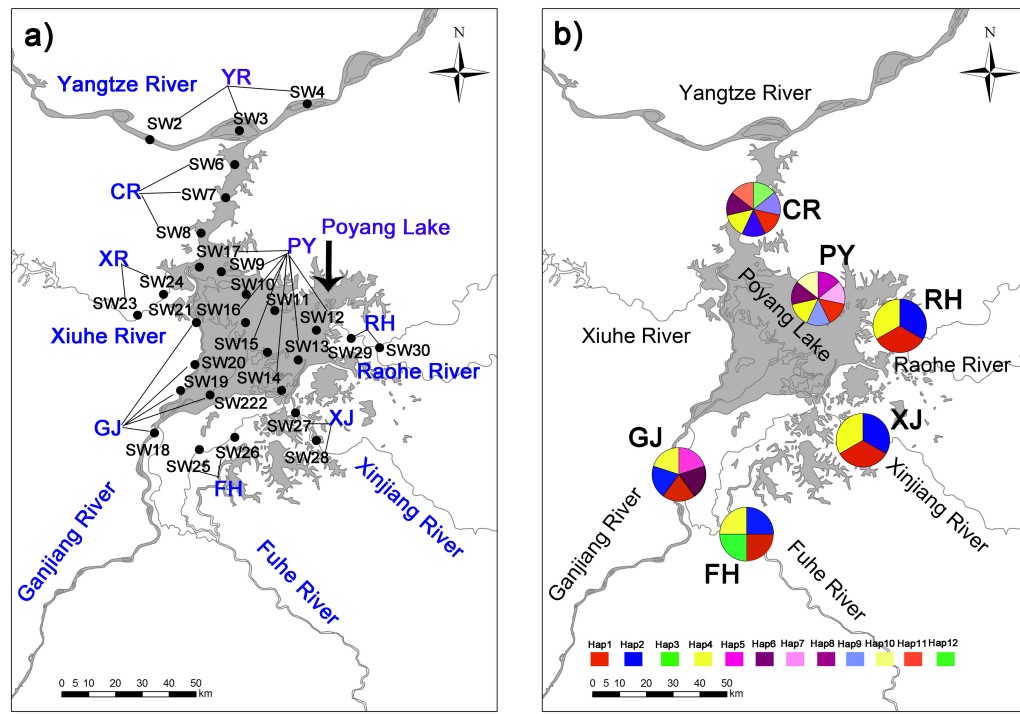

**Figure 1** Collection locations (A) in Poyang Lake Basin with 28 sampling sections three time replicated (84 sampling points not shown for the sake of clarity) where distributed accross eight sampling areas (showed in capital letters), and haplotype distribution (B) of *C. fluminea* in six sampling areas.

pooled together as one sample in order to enrich the data used. In addition, the intercepts of biomass-density relationships were compared between habitats by a unifactorial Analysis of covariance (ANCOVA). The ANCOVA tested for the effect of a predictor (habitat; i.e., lake or peripheral) beyond the effects of one known covariate (density).

## Sample collection and DNA extraction

Samples of *C. fluminea* ($n = 216$) were collected during 2018 and 2019 from the main area of Poyang Lake (PY; 17 specimens), the connected river channel of Poyang Lake (CR; 30 specimens), Ganjiang River (GJ; 30 specimens), Fuhe River (FH; 56 specimens); Xinjiang River (XJ; 44 specimens); and Raohe River (RH; 39 specimens) in Poyang Lake Basin. Tissues of *C. fluminea* samples were preserved in 95% ethanol until DNA extraction. The TINAamp Marine Animals DNA Kit was used to extract the genomic DNA. DNA was evaluated by spectroscopic methods (Nanodrop 2000, Thermo Scientific) and agarose gel electrophoresis.

We used the cytochrome c oxidase subunit-I (COI) primer LCO22me2 (5′-GGTCAACAAAYCATAARGATATTGG-3′) in combination with a reverse primer HCO700DY2 (5′-TCAGGGTGACCAAAAAAYCA-3′) to screen for reliable amplification using the 216 samples of *C. fluminea*. The PCR reaction was performed in a 25-µL volume containing 12.5 µL 2x Taq PCR MasterMix (TianGen), 9.5 µL ddH2O, 1.0 µL forward primer, 1.0 µL reverse primer and 1 µL genomic DNA. PCR amplifications were

performed with an initial denaturation of 94 °C for 2 min, followed by 35 cycles of 94 °C for 1 min, 50 °C for 1 min and 72 °C for 1 min; a final extension temperature of 72 °C for 7 min. Gel electrophoresis was used to confirm successful amplification. An EZ-10 Spin Column PCR Product Purification Kit (Promega, Madison, WI) was used to purify the PCR products. An ABI 3730XL DNA Analyzer (Applied Biosystems, Carlsbad, CA) was used to sequence the purified DNA.

We used DNASP v5.10 (*Librado & Rozas, 2009*) to analyze the number of haplotypes ($H$), haplotype diversity ($H$ d) and nucleotide diversity ($\pi$) for *C. fluminea* population from mtDNA COI sequences. MRBAYES v.3.2.2 (*Ronquist et al., 2012*) was used to construct a phylogenetic analysis using Bayesian inference methods. The fit of twenty-four models of evolution was selected using MRMODELTEST v.2.2 (*Nylander, 2004*). NETWORK 4.5 (*Leigh & Bryant, 2015*) was used to construct a network of COI haplotypes for *C. fluminea* population with a 95% connection limit. An analysis of molecular variance (AMOVA) was used to test the statistical significance of genetic divergence within and among populations and habitats (lake *vs* peripheral) based on ARLEQUIN 3.5 (*Excoffier, Smouse & Quattro, 1992*; *Excoffier & Lischer, 2010*). Neutrality tests and mismatch distribution was used to search for a signature of population expansion. ARLEQUIN 3.5 (*Excoffier & Lischer, 2010*) was used to analyze Tajima's D and Fu's Fs tests.

Mantel tests (Spearman's method; *Legendre & Legendre, 2012*) with 9999 permutations were used to assess the correlations between pairwise water quality parameters matrices and the matrices of density, biomass and genetic diversity. Redundancy analysis (RDA) was used to evaluate variations in density, biomass and genetic diversity in relation to water quality parameters (*Ter Braak & Verdonschot, 1995*). Monte Carlo permutation tests were used with 499 permutations to assess the significance ($P < 0.05$) of the RDA gradient, and the eigenvalues of the first 2 axes were used to measure their importance (*Ter Braak & Verdonschot, 1995*). All water quality parameters, density, biomass and genetic diversity were $\log 10(X + 1)$ transformed to improve their normality before data analysis. CANOCO v 4.5 (*Ter Braak & Verdonschot, 1995*) was used to perform all the ordinations.

## RESULTS

### Commercial harvest and biomass–density relationship

*Corbicula fluminea s.l.* production in Asia increased up to 10,000 tons from 1950 to 1989. Further, it was stable from 1989 until a rapid increase occurred in 2003. Afterwards, production fluctuated between 40,000 and 20,000 tons, with most of this production (ca. 70%) occurring in China.

The total observation number of harvest boats was distributed between 120 and 160 in the Poyang Lake Basin, including lake and peripheral habitats. Lake habitats concentrate more harvest pressure (i.e., 50% of the harvest boats) than peripheral habitats; i.e., half of the harvest boats distributed across the four sites in peripheral habitats. In general, across the six sampling sites within Poyang Lake Basin, the mean harvestable size ± SD for *C. fluminea* was 19.05 ± 4.95 mm ($n = 411$). However, most of the captures (i.e., 74%) were

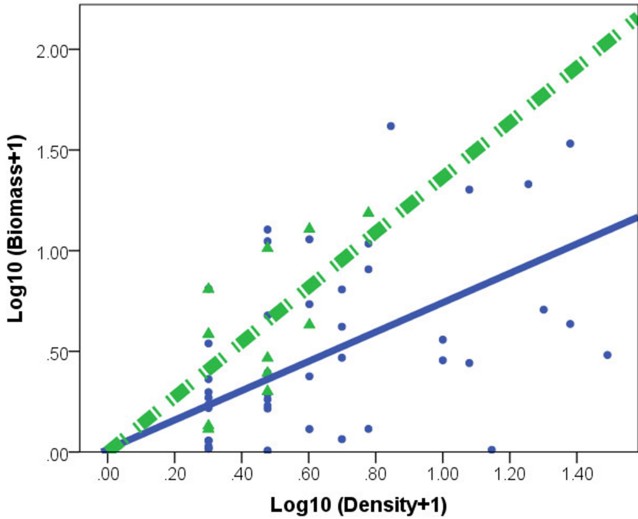

**Figure 2** **Relationships between log biomass and log density for *Corbicula fluminea* in lake (●) and peripheral (▲) habitats.** The dashed line represents the regression line for peripheral habitats. The solid line represents regression line for lake habitats.

≥ 15 mm. There was not a significant effect of habitat on the size of harvested *C. fluminea* [$F_{(1,512)} = 1.288$, $P = 0.257$].

*C. fluminea* density and biomass was higher in lake habitats than in peripheral habitats (Table 1). A significant effect of habitat × density interaction ($F_{(1,73)} = 30.753$, $p = 0.001$) indicated different regression slopes. After accounting for the density effect, the individual of *C. fluminea* from peripheral habitats was heavier and larger than the individual from lake habitats (Fig. 2).

## Population genetics analysis

Twelve mitochondrial COI haplotypes (GenBank accession numbers MN233672-MN233683) were found among 216 sequenced *C. fluminea* individuals around Poyang Lake Basin (Fig. 1). Lake habitats had the greatest variation, with 7 haplotypes (Table 2; Fig. 1). Haplotype diversity ranged from 0.355 to 0.765, and the collection population in Poyang Lake site (PY) had the greatest haplotype diversity. Nucleotide diversity ranged from 0.004 to 0.016, and the collection population Ganjiang River site (GJ) had the greatest nucleotide diversity (Table 2).

The twelve COI haplotype sequences formed two clades: four haplotypes from PY, GJ, CR, FH, XJ and RH (see Table 1 for code correspondences) formed one well supported clade, and eight haplotypes from PY, GJ, CR, FH, XJ and RH formed another clade (Fig. 3). The haplotype network of mitochondrial COI sequences showed that the most frequent haplotype of *C. fluminea* (Hap 4) contained 82 individuals and was shared among individuals from PY, GJ, CR, FH, XJ and RH. The analysis of haplotype network showed similar geographic structure with a general lack of geographic resolution as the phylogenetic analysis (Fig. 3).

**Table 1  Mean ± SD values of *Corbicula fluminea* density (ind m$^{-2}$) and biomass (g m$^{-2}$) among the different habitats, sampling sites and seasons at the Poyang Lake Basin.**

| | | Lake habitats | | Peripheral habitats | | | | | |
| | | Connected river channel of Poyang Lake | Main lake area of Poyang Lake | Yangtze River | Ganjiang River | Xiuhe River | Fuhe River | Xinjiang River | Raohe River |
| | Code | CR | PY | YR | GJ | XH | FH | XJ | RH |
|---|---|---|---|---|---|---|---|---|---|
| Spring | Density | 99.56 ± 63.88 | 54.52 ± 30.22 | 7.11 ± 7.11 | 0 | 0 | 0 | 0 | 0 |
| | Biomass | 15.96 ± 8.27 | 28.52 ± 21.42 | 6.05 ± 6.05 | 0 | 0 | 0 | 0 | 0 |
| Summer | Density | 40.89 ± 40.89 | 15.41 ± 7.02 | 0 | 1.12 ± 1.12 | 0 | 2.78 ± 2.78 | 0 | 0 |
| | Biomass | 58.71 ± 58.71 | 50.24 ± 30.91 | 0 | 3.17 ± 3.17 | 0 | 15.11 ± 15.11 | 0 | 0 |
| Autumn | Density | 37.34 ± 25.20 | 5.93 ± 2.07 | 0 | 0 | 0 | 0 | 17.60 ± 12.04 | 0 |
| | Biomass | 51.63 ± 43.29 | 7.98 ± 3.78 | 0 | 0 | 0 | 0 | 56.39 ± 40.57 | 0 |
| Winter | Density | 0 | 40.89 ± 21.68 | 26.67 ± 10.67 | 6.67 ± 2.72 | 5.56 ± 5.56 | 5.56 ± 5.56 | 0 | 0 |
| | Biomass | 0 | 10.70 ± 7.89 | 16.16 ± 10.95 | 3.99 ± 1.86 | 2.78 ± 2.78 | 25.78 ± 25.78 | 0 | 0 |
| Mean | Density | 44.45 ± 20.57 | 29.19 ± 11.22 | 8.45 ± 6.30 | 1.95 ± 1.60 | 1.39 ± 1.39 | 2.09 ± 1.33 | 4.40 ± 4.40 | 0 |
| | Biomass | 31.58 ± 14.08 | 24.36 ± 9.76 | 5.56 ± 3.81 | 1.79 ± 1.05 | 0.70 ± 0.70 | 10.23 ± 6.29 | 14.10 ± 14.10 | 0 |
**Table 2** Genetic diversity, neutrality tests and mismatch distribution of *Corbicula fluminea* based on mtDNA COI sequences from Poyang Lake Basin.

| Habitat | Collection location | Code | Genetic diversity | | | | Neutrality tests | | | Mismatch distribution | |
|---|---|---|---|---|---|---|---|---|---|---|---|
| | | | *N* | *H* | *Hd* | *π* | Pi (%) | Tajima's *D* | Fu's *Fs* | $P_{SSD}$ | $P_{RAG}$ |
| Lake | Main area of Poyang Lake | PY | 17 | 7 | 0.765 | 0.014 | 7.60 | 0.11 | 2.61 | 0.08 | 0.10 |
| | Connected river channel of Poyang Lake | CR | 30 | 7 | 0.731 | 0.014 | 7.50 | 1.69 | 5.27 | 0.12 | 0.14 |
| Peripheral | Ganjiang River | GJ | 30 | 5 | 0.706 | 0.016 | 8.72 | 2.83 | 9.88 | 0.16 | 0.17 |
| | Xinjiang River | XJ | 44 | 3 | 0.549 | 0.014 | 7.97 | 3.31 | 16.64 | 0.44 | 0.33 |
| | Raohe River | RH | 39 | 3 | 0.355 | 0.007 | 3.82 | −0.17 | 8.37 | 0.06 | 0.37 |
| | Fuhe River | FH | 56 | 4 | 0.421 | 0.004 | 2.26 | −1.31 | 4.04 | 0.24 | 0.26 |
| | Total/Mean | | 216 | 12 | 0.731 | 0.013 | 6.31 | 1.08 | 7.80 | 0.18 | 0.23 |

**Notes.**

*H*, the number of haplotypes; *Hd*, haplotype diversity; *π*, nucleotide diversity; $P_{SSD}$, Sum of Squared deviation *p*-value; $P_{RAG}$, Raggedness *p*-value.

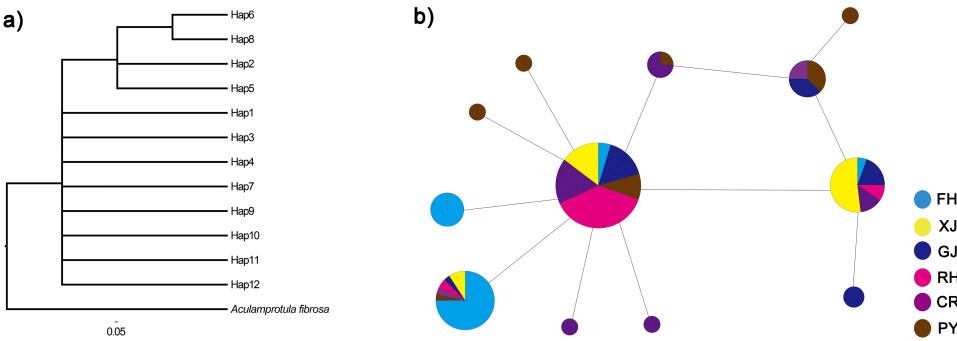

**Figure 3** **Phylogenetic analysis (A) and haplotype network (B) of *C. fluminea* showing the haplotypes identified throughout Poyang Lake Basin sites.** Haplotypes are coloured by region and the size of their circle is proportional to its frequency in the whole sampling effort. Collection population codes are the same as in Table 1.

Based on the analysis of molecular variance (AMOVA), 25.13% of the total genetic variance was found among populations, and the differentiation was significant among populations ($F_{ST} = 0.25$, $p = 0.001$, Table 3). Pairwise $F_{ST}$ among *C. fluminea* populations ranged from 0 to 0.40 with generally significant values (Table 4). The total genetic variance found among peripheral populations was 34.33%, and the differentiation was significant among them ($F_{ST} = 0.34$, $p < 0.0001$). Among lake population, the genetic variance was 1%, but not significant ($F_{ST} = 0.01$, $p = 0.25$). Based on the AMOVA results, a total genetic variance of 0.002% was found between lake and peripheral populations, and the differentiation was not significant between them ($F_{ST} = 0.0001$, $p = 0.47$).There was a lack of significance for the mismatch distribution and neutrality tests based on mitochondrial COI sequences ($p < 0.01$), which indicated that *C. fluminea* population did not experience recently demographic expansion (Table 2).

Table 3 Analysis of molecular variation (AMOVA) calculated using mtDNA COI sequences for *Corbicula fluminea* from six sampling sites in Poyang Lake Basin.

| Source of variation | df | SS | Variance components | Percentage of variation | F-Statistics |
|---|---|---|---|---|---|
| Among populations | 5 | 183.5 | 0.96 | 25.13 | $F_{ST} = 0.25$ |
| Within populations | 210 | 601.9 | 2.87 | 74.87 | |
| Total | 215 | 785.4 | 3.83 | 100 | |

Table 4 Analysis of genetic differentiation coefficient ($F_{ST}$) (below diagonal) and *p*-values (above diagonal) calculated using mtDNA COI among six collection populations of *Corbicula fluminea* from Poyang Lake Basin. Bold font indicates statistical significance ($p < 0.05$). Collection population codes are as in Table 1.

| | PY | CR | GJ | XJ | RH | FH |
|---|---|---|---|---|---|---|
| PY | | 0.004 | 0.003 | 0 | 0.001 | 0 |
| CR | **0.11** | | 0.009 | 0 | 0.002 | 0 |
| GJ | **0.02** | **0.40** | | 0.002 | 0.001 | 0 |
| XJ | **0** | **0.01** | **0.01** | | 0 | 0 |
| RH | **0.01** | **0.02** | **0.01** | **0** | | 0 |
| FH | **0** | **0** | **0** | **0** | **0** | |

## Correlation between density, biomass, genetic and physicochemical parameters

Redundancy analysis showed that 93.6% of cumulative percentage variance of the species–environment relation occurred along the first axis, and along four axes comprised 100% (Fig. 4A). 97.0% of cumulative percentage variance of the species–environment relation occurred along the first axis, and along four axes comprised 100% (Fig. 4B). The density and biomass of *C. fluminea* were correlated with the dissolved oxygen, water velocity, turbidity and water temperature (Fig. 4A). The genetic diversity of *C. fluminea* was correlated with the dissolved oxygen, water velocity, turbidity, pH, salinity and water temperature (Fig. 4B). Therefore, variations in genetic parameters, density and biomass were highly correlated with water velocity ($P < 0.05$; Fig. 4). Moreover, water velocity had a significant effect on the density and genetic diversity of *C. fluminea* based on the Mantel test ($P < 0.05$; Table 5).

# DISCUSSION

## Commercial harvest and biomass–density relationship

*Corbicula fluminea s.l.* is an important economic resource in Asiatic markets. Among them, the Chinese shellfish fishery is the world's biggest producer with about 22,500 tons per year in the period 2003–2017. It should be noted, however, that available information of total *C. fluminea s.l.* production in China is limited by data quality, availability and suitability. For instance, data on *C. fluminea s.l.* aquaculture production from China— which is of relevance for explaining our results—appear for the first time in the official statistics in 2003.

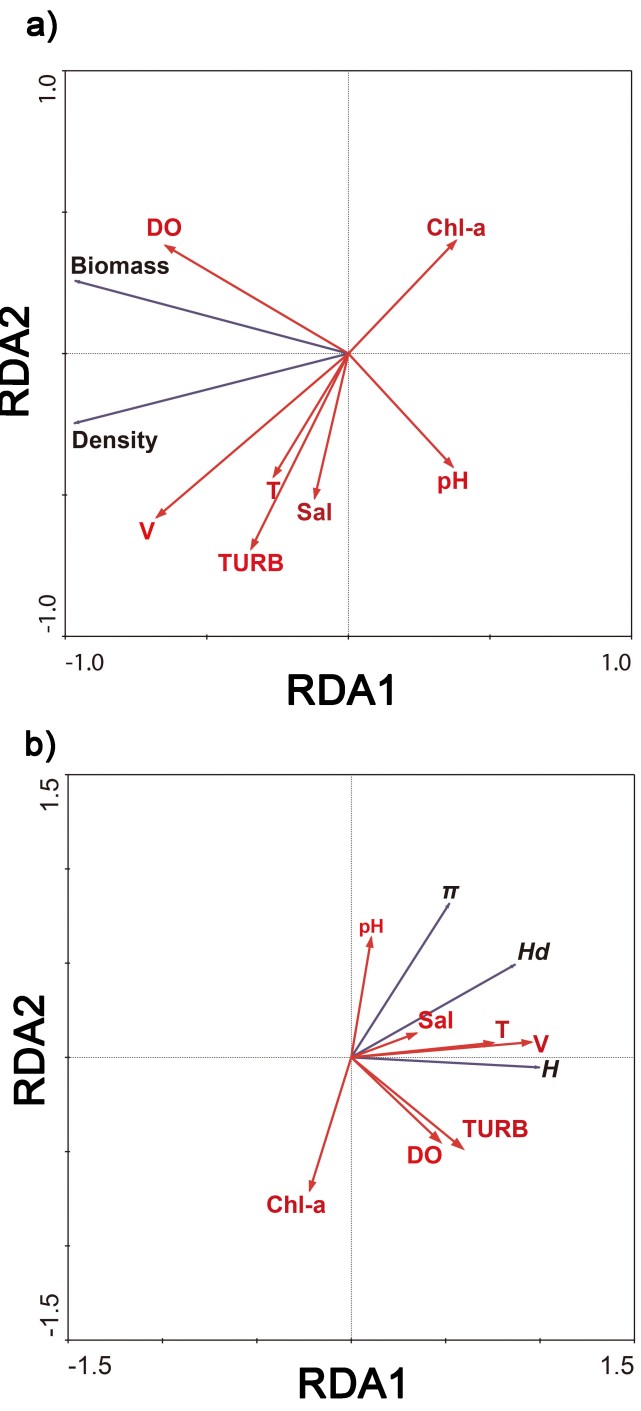

**Figure 4** Ordination biplot of density and biomass (A), genetic diversity (B) and physicochemical parameter obtained by redundancy analysis (RDA) in Poyang Lake Basin.

Here, we discuss demographic and genetic differences among lake and peripheral *C. fluminea s.l.* populations. From a demographic point of view, the harvest data confirm that *C. fluminea s.l.* fishery is focused on large individuals (>15 mm). These harvest data

**Table 5  Effects of physicochemical parameters on density, biomass and genetic parameters obtained from mitochondrial COI in Poyang Lake Basin.** Significant results are shown in bold font ($p < 0.05$).

| | Mantel test | Genetic parameters | | | | D | B |
|---|---|---|---|---|---|---|---|
| | | H | Hd | π | $F_{st}$ | | |
| T | r | 0.250 | 0.301 | −0.069 | −0.373 | −0.236 | −0.196 |
| | p | 0.163 | 0.129 | 0.506 | 0.110 | 0.115 | 0.162 |
| DO | r | −0.012 | −0.018 | 0.199 | 0.072 | 0.209 | 0.272 |
| | p | 0.638 | 0.599 | 0.193 | 0.367 | 0.177 | 0.107 |
| TURB | r | 0.239 | −0.242 | −0.363 | 0.425 | 0.149 | −0.042 |
| | p | 0.167 | 0.163 | 0.083 | 0.133 | 0.243 | 0.465 |
| pH | r | **−0.330** | −0.176 | 0.298 | 0.135 | **−0.321** | −0.215 |
| | p | **0.016** | 0.234 | 0.241 | 0.318 | **0.039** | 0.183 |
| Chl-a | r | −0.195 | 0.137 | **0.670** | −0.300 | −0.214 | −0.239 |
| | p | 0.197 | 0.237 | **0.029** | 0.100 | 0.255 | 0.123 |
| Sal | r | −0.176 | −0.148 | 0.092 | −0.041 | −0.114 | −0.195 |
| | p | 0.327 | 0.362 | 0.412 | 0.501 | 0.556 | 0.170 |
| V | r | **0.865** | **0.617** | 0.042 | −0.048 | **0.419** | 0.191 |
| | p | **0.019** | **0.046** | 0.463 | 0.434 | **0.042** | 0.122 |

**Notes.**
DO, dissolved oxygen; pH, hydrogen ions; TURB, turbidity; T, water temperature; TDS, total dissolved solids; Sal, salinity; Chl-a, chlorophyll-a; V, water velocity.

are necessary to verify our initial assumptions that a fishery such as that described here is phenotypically non-random. In this regard, we hypothesized that harvest can induce demographic and genetic changes in wild *C. fluminea s.l.* populations. Partially supporting this assumption is the observation that the stock of *C. fluminea s.l.* is in decline compared with 20 years ago (Maximum density: 156 ind. m$^{-2}$ in 1997–1999 to 30 ind. m$^{-2}$ in 2016–2017), which reflects the increasing fishing pressure (*Wang et al., 2007*). In adition, our results have shown that lake populations—exposed to intense harvest pressure—have lower biomass at all densities than peripheral populations—exposed to lower harvest pressure. Hence, it is reasonable to assume that higher harvest pressure in lake habitats biased populations towards younger ages and smaller sizes. It should be noted, however, that two quantitative sampling methods (modified Petersen grab and Surber sampler) were used in this study which could potentially bias the study results.

## Effect of human activity on genetics diversity and structure of *C. fluminea*

From a genetic point of view, it is worth noting that Poyang Lake populations share common haplotypes which are also the most common in peripheral populations. This fact strengthens our first hypothesis that populations established in the lake are the source for the peripheral populations. An interesting result, is the presence of six different, some of them unique (refer to Fig. 3B), COI haplotypes in Poyang Lake populations, which could be related with repeated introductions from an alternative source; e.g., via aquaculture production. In general, higher genetic variation resulting from introduced populations allows the species to adapt and spread over large territories (*Roman & Darling, 2007*).

In consequence, it could be expected a greater colonizing potential in *C. fluminea s.l.* population exposed to harvest management.

Even though we find that our results are robust with respect to changes at the population-level, other factors may also account for the population demographic and genetic characteristics. For instance, it should be noted that past drought events related to climate anomalies, water overexploitation or habitat alteration (e.g., Three Gorges Dam) have dramatically affected Poyang Lake biodiversity (*Zhang et al., 2015*; *Wang et al., 2019*). Environmental "bottlenecks" associated to drought conditions may create a new wave front where strong genetic drift may be assumed to be the main mechanism of genetic differentiation among populations (*Peischl & Excoffier, 2015*). In fact, the genetic structure among *C. fluminea* haplotypes was not well resolved based on the phylogenetic analysis of mitochondrial COI sequences. The higher level of genetic differentiation may be also attributed to geographical isolation as an important factor that affects distribution patterns and genetic structure of the species (*Hayes et al., 2008*; *Lv et al., 2013*; *Liu et al., 2017*).

### Effects of environmental changes on density, biomass and genetic parameters

Anthropogenic alterations such as dam construction, water pollution and sand mining at Poyang Lake Basin are important factors responsible of environmental changes (*Xiong, Ouyang & Wu, 2012*; *Jin et al., 2012*). It has been noticed that these kinds of environmental changes may affect aquatic species richness and genetic diversity (*Watson et al., 2016*; *Arroyo-Rodríguez et al., 2013*). In fact, our results indicated that the density, biomass and genetic diversity of *C. fluminea* were correlated with the dissolved oxygen, water velocity, turbidity and water temperature. In addition, these environmental factors have been also related with survival and growth of *C. fluminea* (*McMahon, 1979*; *Karatayev, Burlakova & Padilla, 2005*; *Müller & Baur, 2011*; *Avelar, Neves & Lavrador, 2014*; *Ferreira-Rodríguez & Pardo, 2016*). For example, increase in turbidity may cause siltation potentially increasing mortality rates (*Avelar, Neves & Lavrador, 2014*). Unusually low winter temperatures can result in massive die-offs of *C. fluminea* populations (*Werner & Rothhaupt, 2008*). Increasing temperature is also responsible of increasing metabolic rates but respiration is depleted at temperature up to 30 °C (*McMahon, 1979*). *Müller & Baur (2011)* showed that *C. fluminea* survival in water of 0 °C decreased from 100% to 17.5% with increasing exposure from 4 to 9 weeks. In addition, low river flow and low dissolved oxygen have been related with high mortalities in several macrozoobenthic species, such as *C. fluminea* (*Ilarri et al., 2011*; *Sousa, Antunes & Guilhermino, 2008*).

### Relevance for species management in the invaded range

In addition to commercial harvest, harvest management has been also proposed as an effective strategy for the containment, and long-term control of invasive species (*Pasko & Goldberg, 2014*). Among freshwater animals, *C. fluminea s.l.* has become one of the most successful invasive species all over the world (*Crespo et al., 2015*). The species has measurable negative effects on native species and ecosystems (e.g., competition with

native mollusks and habitat alteration; *Ferreira-Rodríguez et al., 2018*; *Ferreira-Rodríguez, Iglesias & Pardo, 2019*). On this regard, studying invaders in their native range can shed some light in the complex array of interactions that determine its successful establishment and management opportunities in other world regions (*Hierro, Maron & Callaway, 2005*). Nevertheless, prior the establishment of any management action, there must be a reasonable prospect that the risk of re-invasion is as low as possible. In this regard, haplotype differentiation may help to identify isolated populations where control or eradication measures will be implemented.

## CONCLUSIONS

Present results have important implications in the native and invaded range. First, in the native range, *C. fluminea s.l.* supports an important fishing industry. In the particular case of the Poyang Lake Basin, the number of harvest boats has been estimated on the basis of personal unstructured observations which do not give us perfect answers, but it gives an estimate to characterize this particular fishery and how lake and peripheral habitats differ in demographic and genetic characteristics. In this regard, lake populations exposed to higher harvest pressure showed demographic and genetic differentiation. Hence, it is likely to think that harvest management may affect population parameters in some degree. These results are especially relevant for those species of economic value because sustainable management should maintain populations at or above viable levels. In addition, our results have shown that harvest management can reduce population numbers. It should be noted, however, that harvest management as control strategy can be a high risk undertaking because it could incentivize deliberate long-distance introductions increasing the genetic diversity and, in consequence, enhancing its invasive potential.

## ACKNOWLEDGEMENTS

We are very grateful to Dr. Emilio Rolán for his insightful comments. The authors alone are responsible for the content and writing of this article.

### Funding

This work was supported by the National Key R & D Program of China (2018YFD0900801) and Noé Ferreira-Rodríguez was supported by a post-doctoral fellowship (Xunta de Galicia Plan I2C 2017-2020, 09.40.561B.444.0) from the government of the autonomous community of Galicia. The funders had no role in study design, data collection and analysis, decision to publish, or preparation of the manuscript.

### Grant Disclosures

The following grant information was disclosed by the authors:
National Key R & D Program of China: 2018YFD0900801.
Xunta de Galicia Plan I2C 2017-2020: 09.40.561B.444.0.

## Competing Interests

The authors declare there are no competing interests.

## Author Contributions

- Weikai Wang and Xiongjun Liu conceived and designed the experiments, performed the experiments, analyzed the data, prepared figures and/or tables, authored or reviewed drafts of the paper, and approved the final draft.
- Noé Ferreira-Rodríguez conceived and designed the experiments, analyzed the data, prepared figures and/or tables, authored or reviewed drafts of the paper, and approved the final draft.
- Weiwei Sun and Yanli Wu conceived and designed the experiments, performed the experiments, prepared figures and/or tables, and approved the final draft.
- Shan Ouyang, Chunhua Zhou and Xiaoping Wu conceived and designed the experiments, prepared figures and/or tables, authored or reviewed drafts of the paper, and approved the final draft.

## DNA Deposition

The following information was supplied regarding the deposition of DNA sequences:
DNA sequence data are available at NCBI: MN233672-MN233683.

## Data Availability

The raw measurements and raw data are available in the Supplemental Files.

## Supplemental Information

Supplemental information for this article can be found online at http://dx.doi.org/10.7717/peerj.9657#supplemental-information.

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
