# Peer review of "Demographic and genetic characterization of harvested Corbicula fluminea populations"

_PeerJ, doi:10.7717/peerj.9657_

## Round 0.1 · original submission · Major Revisions

Your manuscript has to be deeply improved. There are many sentences expression and some grammar errors, so the MS has to be revised and corrected by a native English speaker.

The genetic analysis presented is not enough: too few specimens were studied, unclear from which year/month - and just a single marker was considered.

The comparisons between demographic and genetic variables have to be properly carried out.

·

Basic reporting

The use of English language needs moderate improvement, as there are numerous small spelling and grammar mistakes. Here are just a few examples:

Line 79: change "in" to "on"

Line 184: Delete "mo". On the next line, change "occurred" to "occurring".

Line 308: change "poptential" to "potential"

Experimental design

I thought the research question was not well defined. The study has basically taken genetic and demographic data from a number of locations and performed some analyses providing information on spatial variation, but the introduction and discussion text mentions many research questions only loosely related to these data and analyses:

For example, there are no data presented here from the invaded range, so these data are not closely relevant to research questions about invasion biology. Any connection between these data and invasion biology could be limited to a few sentences of speculation in the discussion.

It is stated that “The goal of this study was to test whether commercial harvest will affect demographics and genetics of C. fluminea”, but this was not really tested, or only very indirectly. Comparisons were done among "lake" habitats and "peripheral" habitats, and basic observations suggested fishing pressure differed between the habitats. However, the habitat-type, and sampling methods used, also differed between lake and periphery sites, so it is unknown if the demographic and genetic differences are a result of fishing, habitat-type, or sampling method. From Fig. 1, it seems only two sites were in the fished lake area and four were in the peripheral area, so its difficult to tell if any differences may also just be random spatial variation. A more complex experimental design than was used here is needed to answer the question of “whether commercial harvest will affect demographics and genetics of C. fluminea”.

There is mention of harvest data and numbers of dredging boats, but these data do not seem to be collected very systematically or comprehensively, are only mentioned briefly in the results, and are not analysed. If these data are to be included, they should be expanded upon, e.g. displayed graphically and analysed.

The authors aim in many cases to compare demographic and genetic variables between “lake” and “peripheral” habitats, but these comparisons are all done informally, i.e. no direct analysis involving lake/periphery as a distinct factor was done. Possibly the authors could re-analyse the data and attempt to use their lake and periphery sites or sections as replicates in direct comparisons (e.g. ANCOVA).

Validity of the findings

As mentioned above, there are considerable amounts of data collected here, but I do not think it is very robust for testing the stated aims.

Concerning the conclusions, I do not think these analyses can be used to make strong conclusions about harvest management or colonisation potential in invaded ranges, or about non-random harvest being a potential driver for population demographic and genetic characteristics. The main conclusions that can be derived from the current analyses are about spatial variation in demographic and genetic characteristics, which only may be influenced by harvesting. Possibly with some re-analysis of the data the link of the analyses with conclusions about harvesting can be made stronger. Or possibly the conclusions could be changed to better reflect the limitations of the current analyses.

Additional comments

Line 111: I am not sure that the presence of dredging boats can be used as a proxy for fishing pressure – e.g. maybe the boats were observed just passing by or doing other activities besides fishing.

Line 116: Figure 1 does not show these 84 sampling points and 28 sampling sections and eight sampling areas. The terms “sites”, “points”, “sections” and “areas” have been used without many details, which is confusing. Please provide more details on how these points, sections, sites and areas were arranged.

Line 121: The same sampling methods should have been used across all sites, not a grab sampler at one type of site and a net sampler at another. One main comparison that was attempted was between lake and peripheral sites, but as they used different methods, maybe the differences observed were simply because of the different methodologies.

Line 134: There is not much mention of these environmental data analyses in the results, please describe more in the results how the environmental variables are correlated with the demographic or genetic data. If the authors are interested, it would be my recommendation that in a re-write of this manuscript, the association of the environmental and biological data could be given greater emphasis, to make the conclusions more valid and the manuscript more interesting to a broader audience.

Line 188: Perhaps it would be better to present the fishing boat data as density, i.e. boats per km2?

Line 195-196: I think this sentence needs to be rephrased, as Figure 2 shows the highest biomass samples were from the lake habitats, and Table 1 shows the mean was also higher there.

Line 200-222: I had trouble understanding how this sentence relates to the figure 4, please provide more detailed results explanation here so the link between demographic expansion and frequencies of pairwise differences in Fig. 4 is more clear.

Line 239-240: Harvesting any individuals will reduce stock abundances, not just harvesting large ones, so this sentence does not seem to be very relevant to the hypothesis mentioned in the preceding sentence.

Overall, I found the discussion a bit difficult to read, with the subject matter changing unexpectedly and the links from one sentence to the next often not clear. Also, the links of much of the discussion text with the actual data collected in the study are also not clear. I would suggest a thorough re-write, ideally structured around some modified or extra analyses.

Line 263-265: This seems a bit obvious - larger individuals are being harvested, so you end up with populations with smaller sizes remaining. I do not think all these complex and confusing ideas about self-regulated intra-specific competition or running water affected peripheral population food delivery need to be emphasised or probably even mentioned, especially given that the data are only loosely linked to direct evidence of fishing pressure, and not closely linked at all to competition or food delivery.

Line 299: I think the world "innumerable" is an exaggeration and should be removed.

Line 303: Much of the introduction and earlier discussion talks about how data are needed to learn about how harvesting may be used to control this clam in the invaded range, but here suddenly it is stated that this should not be considered because it could "incentive deliberate long-distance introductions". This should be stated up front (e.g. in the introduction) as a reason to not consider this concept.

Figure 1: please make it more clear here which of these locations are the main lake ones and which are the peripheral ones.

Reviewer 2 ·

Basic reporting

no comment

Experimental design

no comment

Validity of the findings

no comment

Additional comments

I have read and review the manuscript carefully entitled “Demographic and genetic changes in wild Corbicula fluminea populations under harvest management”. The manuscript investigated the density, biomass and genetic structure of Corbicula fluminea based on CO1 gene in Poyang Lake Basin (lake and peripheral habitats). The meaning of this manuscript is significant, but there are still many problems in the whole article due to the lack of data analysis directly related to harvest pressure (or harvest management). In a whole, I am inclined to accept this paper after a major revision or rewriting. The general comments and specific comments are as follows:

General comments:
1) The abstract section gave a lot of explanations for the reason and significance of the research, but it didn't show the main results of the study, just a few conclusions. and the same happened in the conclusion section. Rewrite is recommended. I suggest that the author rewrite these sections.
2) This manuscript hope to study the population and genetic changes of Corbicula fluminea under harvest pressure in the Poyang Lake Basin. But the results only described the production changes in Asia and China, and the general distribution of harvesting boats in the Poyang Lake Basin. There was no direct correlation between harvest pressure (or harvest management) and population changes (or genetic changes). Without the support of directly related data analysis, it is obviously insufficient to discuss the main scientific issues of this manuscript, so it is suggested to add correlation analysis.
3) The sections of introduction, discussion and conclusion contained a lot of content for Corbicula fluminea invasion, but the study was a native range research. There was no data analysis and research results about Corbicula fluminea invasion. Although this study had certain significance for Corbicula fluminea invasion, but the meaning was limited, and the relevance to the major scientific issues was weak. If the author wants to discuss this problem, please add relevant research analysis, otherwise you need to weaken this issue.
4) There were too many indirect and speculative discussions in the discussion section, and there were no directly related analysis results to support the discussions. Therefore, it is suggested that the discussion should be rewritten and discussed based on the directly related results.
5) The result for figure 5 showed that C. fluminea has had a relatively small recent demographic expansion event occurring between 1,000 -2,500 years ago. This is a large-scale issue, which may be related to geological events or climate change, and this is another scientific issue, which is less relevant to the main research issue of this paper. Suggest to delete.
6) Last but not least, there are many sentences expression and some grammar errors, so I urge the authors to polish the MS by a native English speaker.

Some specific comments are as follows:
1) When the comparative degree of adjectives is used (such as line 55), the point of comparison should be clearly indicated. If the comparison is unclear, use the positive degree of adjectives. Please check whether there is any such problem in the full text.
2) Line 92: Please change " near threatened with extinction" to " be endangered"
3) Line 132 -142: The methods section had an introduction to the methods for obtaining and analyzing the environmental data, but the results section had not any content of the environmental results. This is not appropriate. Suggest to add environmental data results or delete environmental data-related methods.
4) Line 143-144: “No significant differences … p<0.05; results”: My understanding is that p < 0.05 is significant, which is inconsistent with the result that there is no significance in the text. Please check.
5) Line 192: “Further, it … increase occurred.”: Confused. Suggest change to “Further, it was stable from 1989 until a rapid increase occurred in 2003.”
6) Line 195-196: “The total number … lake and peripheral habitats.”: The one observation shown here or the total observation? Please clear.
7) Line 196-198: “Lake habitats … dredging boats… in peripheral habitats.”: Dredging boats here should be referring to the harvest boats of Corbicula fluminea, rather than sand dredger, are they? This needs to be clear in the methods.
8) Line 205-207: “After accounting for … higher than biomass from lake habitats”: I don't think the results here is accurate. I think figure 2 shows that the individual of C. fluminea from peripheral habitats was heavier or larger than the individual from lake habitats.
9) Line 226: “FST =0.25”: Here, FST =0.25 refers to the value among populations, while in table 3, it refers to the value within populations. Please check it.
10) Line 250-255: “In this regard, we hypothesized … reflects the increasing fishing pressure (Wang et al., 2007).”: There were too many possible reasons for the sharp drop in density, and there was no relevant data analysis. Only due to large individual have been harvested, it is not convincing to infer that the decline in density was caused by fishery pressure. Suggest adding some valid direct data analysis to support this discussion.
11) Line 268-271: There were no relevant environmental data results in the results section, but there's a related discussion going on here. Suggest to add this part of the results.
12) Line 283-284 and line 301-302: Line 283-284 showed low differentiation among populations, and Line 301-302 showed high differentiation among populations. These two conclusions seemed to be contradictory. Why did these conclusions occur? Please explain and discuss.
13) Line 287-289: Please add a reference at the end of the sentence “In general, higher genetic variation resulting from introduced populations allows the species to adapt and spread over large territories.”
14) Line 298-301: Discuss why this is the case. Were there not enough samples or something else?
15) In the conclusion section, please summarize the conclusions directly related to the research results and suggest rewriting.
16) In figure 3, there are two yellow parts in Hap1, please check for drawing errors.

Annotated reviews are not available for download in order to protect the identity of reviewers who chose to remain anonymous.

Reviewer 3 ·

Basic reporting

Mostly very clear and unambiguous, professional English. Literature references dominated by phytogeographic and ecological citations, few on state-of-the-are population genetics.
Article stucture good. Statistical phytogeography results (Figs. 4 and 3 - btw they are swapped) interesting but less relevant for the question of the study.
Results are relevant but cannot answer hypotheses 2 on genetic diversity change under harvest pressure.

Experimental design

Questions well-defined and relevant. Sampling very good, genetic analysis: to few specimens, unclear from which year / month - and just a single marker. This is inappropriate for such an important question that aims to guide conservation managment.

Validity of the findings

Conclusions on part 2 are not supported (population genetic analysis) as only based on few specimens per sampling (when considering the 84 samplings 216 individuals is not much). The questions is really interesting and relevant - how strong is the impact of harvesting of populations, yet a traditional phylogeographic single-marker analysis will not be of great help here; it provides very interesting first insights thought.
Conclusions on part 1 (smaller individuals under harvest pressure) is well supported but totally expected.

Additional comments

The study by Wang et al. present interesting small-scale results into harvest pressure on C. fluminea populations in the lake and the rivers connecting to it. Under strong harvest pressure, sizes of specimens are smaller as predicted. A clear strength of the study is the effort used to assess biomass x density information across years and sites. Very good data, yet it needs to be shown in a better way. The second and main part of the study addressed the question if different harvest pressures impact on population genetic diversity. Here the study presents haplotype information of different populations around Poyang Lake and suggests that no impact of harvesting on genetic diversity is found. While this may be true, the population genetic analyses are no state of the art as
i) the sample size for genetic analysis is rather small considering the 84 sampling from different time points at sites, and
ii) the whole analysis is based on a single, mitochondrial marker. It is especially the latter that clearly limits the main results of harvest pressure.
True, in the late 1990s and early 2000s single-marker analyses for population genetic inferences were often done, but it has often been shown in conservation genetic research that single marker, especially mtDNA marker analyses can be misleading. Therefore, I cannot reccommend publication as genetic results (second part of analysis) has too little explanatory power. Also, mismatch analyses, skyline plots - all these are phylgenetic methods that will assess effects over thousands of years and not recent harvest-specific impacts. My recommendation is to add a few (e.g. 10) microsatellites (from Penarrubia et al. 2015 e.g.) and reanalyse the data, use proper population genetic statistics based on heterozygosity, allelic diversity. I would also recommend taking more from what you have - you have a temporal data set. Why not assess population parameters (haplotype diversity, allelic richness, heterozygosity) per site per sampling time point. With that you would be able to infer effective population size parameters - a much needed are rarely provided esimator for managers.

Specific comments:
- It is interesting that for 84 sampling points only 216 haplotypes were analysed - why not more? This study has an immense potential by studying genetic change over time (as you have the different time points) to also infer effective population size. What changed in populations between years? This effect can be translated to Ne using the temporal methods given in the literature.

- Better outline, which samples where obtained in which year - and what was the sample size. Perform a rarefaction analysis to see if haplotype diversity per site is saturated (take the different sampling time points as your replicates)

- Check this study for available microatellites: Peñarrubia L, Araguas RM, Pla C, Sanz N, Viñas J, Vidal O (2015) Identification of 246 microsatellites in the Asiatic clam (Corbicula fluminea). Conserv Genet Resour 7: 393–395.)

- L98: Wrong defintition of genetic drift. You mean drifting organisms. Genetic drift is random fluctuation of alleles

---

## Round 0.2 · Minor Revisions

Your manuscript has been deeply improved although still needs some minor corrections, in order to correct some mistakes or to clarify some concepts. Please, consider all the reviewers' suggestions in the revised manuscript.

·

Basic reporting

The use of English language is improved. There were a small number of additional corrections needed:

line 77: change to "actually"

line 79: change to "Because of the"

line 185-186: change to "harvest boats were distributed"

line 239: change to "such as that described"

line 274: change to "anthropogenic"

line 276: change to "that these kinds"

line 296-297: change to "differentiation" and to "isolated".

line 309: change to "incentivize"

Figure 1 caption: change "accross" to "across"

Experimental design

no comment

Validity of the findings

no comment

Additional comments

line 36: please proved a few more details in the abstract what exactly a "peripheral" location is, so readers can understand this without having to go into the main text. On the next line it is stated that differences of the clams may be caused by harvesting, please make the link here between lake and peripheral locations, and harvesting pressure. Overall, the abstract is short with few details, it could be expanded with more sentences stating in-depth details of main results and discussion.

line 37-39: Please provide some more details about why the consequences of demographic and genetic changes presented here are especially relevant for harvest management and sustainability of the resource.

line 40: Disturbance is not measured in this study so I do not think "Disturbance ecology" should be a keyword. Also, this study is only of a native species within its native range so I do not think "Alien invasive species" should be a keyword.

line 86: most readers will not know what a "peripheral" population is. As it stands, the reader has to guess that this means the areas "surrounded by modified inland artisanal aquaculture systems" mentioned above. Please describe exactly what a "peripheral" areas is here and how it differs from a "lake" area. Also on Figure 1 it does not specify anywhere which habitats are "lake" and which are "peripheral". There are some areas marked "CR" in this figure - are they peripheral or lake? All other abbreviations are explained on this figure, please explain what "CR" stands for as well.

line 111: please provide some details or a reference about what a Surber sampler is.

line 248: change to "two quantitative sampling methods (modified Petersen grab and Surber sampler) were used"

line 290-291: please provide a reference for this statement.

line 296-297: the concept introduced in this sentence is a bit unclear, I ask the authors to expand upon this concept a bit to make it easier for the reader to understand.

Reviewer 2 ·

Basic reporting

I have received the revised manuscript entitled “Demographic and genetic changes in wild Corbicula fluminea populations under harvest management” and have carefully reviewed it. The authors have rewritten the main question in this manuscript. The manuscript got a great improvement over the first draft. However, there are still some problems with the manuscript, and I suggest a further revision. The comments are as follows:

1) Since there was no harvest data available for direct correlation analysis with C. fluminea population, and the authors also have revised the main scientific question of the manuscript to weaken the issue of harvesting pressures, so I suggest that this issue should also be weakened in the title, abstract, goal and conclusion.
2) The PDF version of the abstract is inconsistent with the Word version. For example, line 35: “We found higher biomass…” in PDF version, but “We found lower biomass…” in Word version. Please check.
3) Line 115-122 “Environmental data were measured in … were referenced from Li et al. (2019).”: Are physical and chemical parameters measured by yourselves or by reference? It's confusing.
4) There is Principal component analysis (PCA) introduction in the Methods section, but there is no relevant result in the Result section. Please check. In addition, in the Methods section, statistical analysis methods of data should be put together and subtitled. Please reorganize the Methods section.
5) There is RH region but no XH in Figure 3b, but there is XH on the corresponding line 204-207. In addition, haplotype annotations are missing in Figure 3b.
6) It is recommended to provide R2 and p values associated with RDA analysis to understand the correlation significance and degree of environmental factors. The discussion on the effects of environmental factors is not sufficient. It is suggested to have an in-depth discussion based on the results of RDA.

Experimental design

no comment

Validity of the findings

no comment

Additional comments

I have received the revised manuscript entitled “Demographic and genetic changes in wild Corbicula fluminea populations under harvest management” and have carefully reviewed it. The authors have rewritten the main question in this manuscript. The manuscript got a great improvement over the first draft. However, there are still some problems with the manuscript, and I suggest a further revision. The comments are as follows:

1) Since there was no harvest data available for direct correlation analysis with C. fluminea population, and the authors also have revised the main scientific question of the manuscript to weaken the issue of harvesting pressures, so I suggest that this issue should also be weakened in the title, abstract, goal and conclusion.
2) The PDF version of the abstract is inconsistent with the Word version. For example, line 35: “We found higher biomass…” in PDF version, but “We found lower biomass…” in Word version. Please check.
3) Line 115-122 “Environmental data were measured in … were referenced from Li et al. (2019).”: Are physical and chemical parameters measured by yourselves or by reference? It's confusing.
4) There is Principal component analysis (PCA) introduction in the Methods section, but there is no relevant result in the Result section. Please check. In addition, in the Methods section, statistical analysis methods of data should be put together and subtitled. Please reorganize the Methods section.
5) There is RH region but no XH in Figure 3b, but there is XH on the corresponding line 204-207. In addition, haplotype annotations are missing in Figure 3b.
6) It is recommended to provide R2 and p values associated with RDA analysis to understand the correlation significance and degree of environmental factors. The discussion on the effects of environmental factors is not sufficient. It is suggested to have an in-depth discussion based on the results of RDA.

---

## Round 0.3 · accepted · Accept

Thank you very much for improving your manuscript and thank you also for submitting your work to this journal.